# Procapra Przewalskii Tracking Autonomous Unmanned Aerial Vehicle Based on Improved Long and Short-Term Memory Kalman Filters

**DOI:** 10.3390/s23083948

**Published:** 2023-04-13

**Authors:** Wei Luo, Yongxiang Zhao, Quanqin Shao, Xiaoliang Li, Dongliang Wang, Tongzuo Zhang, Fei Liu, Longfang Duan, Yuejun He, Yancang Wang, Guoqing Zhang, Xinghui Wang, Zhongde Yu

**Affiliations:** 1North China Institute of Aerospace Engineering, Langfang 065000, China; 2Key Laboratory of Land Surface Pattern and Simulation, Institute of Geographic Sciences and Natural Resources Research, Chinese Academy of Sciences, Beijing 100101, China; 3Aerospace Remote Sensing Information Processing and Application Collaborative Innovation Center of Hebei Province, Langfang 065000, China; 4National Joint Engineering Research Center of Space Remote Sensing Information Application Technology, Langfang 065000, China; 5Agricultural Information Institute of Chinese Academy of Agricultural Sciences, Key Laboratory of Agricultural Monitoring and Early Warning Technology, Ministry of Agriculture and Rural Affairs, Beijing 100081, China; 6University of Chinese Academy of Sciences, Beijing 101407, China; 7Key Laboratory of Adaptation and Evolution of Plateau Biota, Northwest Institute of Plateau Biology, Chinese Academy of Sciences, Xining 810001, China; 8Intelligent Garden and Ecohealth Laboratory (iGE), College of Biosystems Engineering and Food Science, Zhejiang University, 866 Yuhangtang Road, Hangzhou 310058, China

**Keywords:** Procapra przewalskii protection, autonomous unmanned aerial vehicle, object tracking, Kalman filter, long and short-term memory

## Abstract

This paper presents an autonomous unmanned-aerial-vehicle (UAV) tracking system based on an improved long and short-term memory (LSTM) Kalman filter (KF) model. The system can estimate the three-dimensional (3D) attitude and precisely track the target object without manual intervention. Specifically, the YOLOX algorithm is employed to track and recognize the target object, which is then combined with the improved KF model for precise tracking and recognition. In the LSTM-KF model, three different LSTM networks (*f*, *Q*, and *R*) are adopted to model a nonlinear transfer function to enable the model to learn rich and dynamic Kalman components from the data. The experimental results disclose that the improved LSTM-KF model exhibits higher recognition accuracy than the standard LSTM and the independent KF model. It verifies the robustness, effectiveness, and reliability of the autonomous UAV tracking system based on the improved LSTM-KF model in object recognition and tracking and 3D attitude estimation.

## 1. Introduction

Procapra przewalskii is an endangered ungulate endemic to the Qinghai-Tibet Plateau. Its type specimen was collected by Nikolai M. Przewalski in the Ordos Plateau of Inner Mongolia in 1875 [1]. It is listed as a national key protected wild animal by the *List of National Key Protected Wild Animals* issued in 1988. In 2001, it was listed as one of the 15 species in urgent need of rescue in the *National Wildlife Conservation and Nature Reserve Construction Project Master Plan*. In December 2002, the State Forestry Administration formulated the *Overall Plan of the National Proctor* for the gazelle protection project. Meanwhile, it was determined to be a critically endangered (CE) species by the Red List of the International Union for Conservation of Nature (IUCN) in 1996 and 2003. Due to the recovery of local populations and the discovery of new populations, it was adjusted as endangered (EN) from CE by the Red List of the IUCN, but it was still listed as CE in the assessment results of the Red List of Biodiversity in China [2]. Perennial cold and oxygen deficiency in plateau areas pose more difficulty in continuous artificial monitoring of animal behaviors.

As a non-contact data acquisition method, unmanned aerial vehicles (UAVs) exhibit unique advantages in wildlife monitoring. Thanks to the significant progress in computer vision technologies, autonomous UAVs equipped with edge reasoning units have been applied in preliminary real-time monitoring [3,4]. UAVs are limited by size, weight, and power (SWaP), and a single camera is usually equipped as the best sensor. However, when the single-shot attitude estimation is applied to video data, it has to estimate the extraordinary noise and confuse the visually similar but spatially different image features if the time information is ignored. Therefore, a time filter is an effective way to increase the accuracy of attitude estimation. The Kalman filter (KF) [5] is a very wide choice due to its simplicity and versatility. In addition, the extended KF (EKF) [6] can address the nonlinear systems of measurement and transition models.

However, these measurement and transition models cannot be specified a priori, severely restricting the application of KFs. Therefore, they can try to directly learn the motion models from training data with support vector machines [7] or with long and short-term memory (LSTM) [8] to overcome the above limitations. It can release the modeler from time-consuming selection and optimization of the KF by learning the motion models, simultaneously enriching the underlying models. However, sufficient training data should be ensured to cover all possible motion paths of the tracked object when learned motion models are applied to enforce temporal consistency in pose estimation.

This work mainly contributes to an autonomous UAV target-tracking system based on the improved LSTM-KF model. (1) A three-dimensional (3D) attitude tracking algorithm integrating a learning-based real-time target detection algorithm with a UAV embedded system based on target detection, stereo reconstruction technology, and the KF model, is implemented in the low-cost UAV system to automatically identify, locate, and track the Procapra przewalskii. (2) In the LSTM-KF model, the calculation formulas of the output gate *Q_t_* of three different standard LSTM networks are modified by connecting them with the input gate to solve overfitting during the training of many datasets about the Procapra przewalskii video sequences. (3) During the training iteration of the LSTM-KF model, the Adam optimizer is modified by extending the update rules based on the *L^2^* norm in Adam optimization to those based on the *L^p^* norm. The improved model is able to accurately track the behavioral attitudes of the LSTM-KF model to identify the original model and reduce the modeler’s a priori burden to specify motion and noise models.

This paper is structured as follows. In the “Introduction” section, the background and definitions of Procapra przewalskii, autonomous UAV object-tracking system, KF, and LSTM model are introduced. In the “Related Works” section, the background of the CNN-based animal-monitoring algorithm and the learning-based KF structure are described. The “Materials and Methods” section describes the hardware system architecture used for UAV object tracking, the study area and dataset, the definition and principles of animal 3D pose estimation, the YOLOX model, and the structure of the improved LSTM-KF model. Next, the experimental results are introduced and analyzed. The performances of the YOLOX model at different resolutions are compared, and the tracking results and average errors of the improved LSTM-KF model are verified. In addition, the tracking effects of Procapra przewalskii based on simulation systems and actual flight scenarios are presented. In the “Conclusions” section, the results and discussions are summarized, and future research directions are prospected.

## 2. Related Works

### 2.1. Animal-Monitoring Algorithm Based on Convolutional Neural Network (CNN)

CNN-based methods that can achieve accurate result-oriented image feature extraction and representation have been increasingly accepted and applied in vision and image processing [9,10,11]. Meanwhile, they have exhibited extensive applications in animal identification without pre-specifying any features [12] but have limited usages in monitoring the activities of farming animals [13]. The two-stream network proposed by [14] is one of the tracking models to track moving objects. Multi-layers and the optical flow of convolutional networks enable the capturing the frames of relevant information on an object and tracking the object movement among various frames. Shortly after the proposal of two-stream networks, long-term recurrent convolutional networks (LRCNs) were developed. LRCNs [15] generally comprise several CNNs, namely Inception modules, ResNet, VGG, and Xception, enabling the extraction of spatial and temporal features.

LRCN was the most applied tracking model thanks to its reasonable architecture in object tracking. Generic object tracking using regression network (GOTURN) [16] is another lightweight network that achieves 100 frames per second (fps) for object tracking. GOTURN was initially trained with the generic-objects-filled datasets. The regions of interest (ROIs) on the frames are undertaken as input data for the trained network, providing the possibility of continuously predicting the location of the target. The Slow Fast network [17], on the other hand, tracks objects using two streams of frames, namely slow and high pathways. Many other algorithms can be introduced for animal monitoring, including, but not limited to, simple online and real-time tracking (SORT), the Hungarian algorithm (HA), the Munkres variant of the Hungarian assignment algorithm (MVHAA), the spatial-aware temporal response filter (STRF), and the channel and spatial reliability discriminative correlation filter (CSRDCF) [18,19,20,21].

These algorithms were utilized by object-detection models, such as Faster R-CNN, FCN, SSD, VGG, and YOLO, to detect and track animals in images using their geometric features in continuous frames [13]. To provide a reliable and efficient method to monitor the behavioral activity in cows, [22] presented a tracking system embedded with ultra-wideband technology. Similarly, [23] employed a computer vision module to analyze and detect the positive and negative social interactions in feeding behavior among cows. The system was implemented and tested on seven dairy cows in the feeding area, realized an accuracy with a mean error of 0.39 m and standard deviation of 0.62 m, and achieved a detection accuracy of social interactions of 93.2%. However, the real-time locating system (RTLS) exhibits poor accuracy in identifying individual cows if they are in close body contact.

The CNN-based algorithm presents limited applications in monitoring agricultural animals because it does not pre-specify any features of any target. In this case, the YOLOX model, a lightweight network with an anchorless frame at the head of the network, is applied in this paper, which is equipped with several high-performance detectors and a faster network convergence, so that animals with specified features can be monitored. LRCN is the most widely used tracking model due to its reasonable structure in target tracking. However, it fails to effectively monitor the pose behavior of animals and consider the variations in measurement noise. Therefore, the LSTM-KF model with good robustness, reliability, and validity in target tracking and 3D pose estimation is proposed in this paper, which can alleviate the effect of measurement noise and modify the measurement results.

### 2.2. Learning-Based KF Architecture

In the current work, machine learning and KF models are combined for temporal regularization. The approaches can be classified into those that learn the static parameters of the KF and those that actively regress the parameters during filtering. The noise covariance matrices (NCMs) were optimized statically by [24] to replace the manual fine-tuning of noise parameters in a robotic navigation. Additionally, a coordinate ascent algorithm was employed and each element of the NCM was optimized. However, this approach is only applicable for noisy but time-invariant systems. As opposed to the dynamic model adopted in this study, a change in measurement noise cannot be considered and will therefore lower the accuracy in state estimates.

Reference [8] learned the underlying state transition function that controlled the dynamics of a hidden process state. However, only the state space equations of the KF are used instead of the prediction and update scheme with good performance under linear state transitions and additive Gaussian noise [6]. The neural network models that jointly learn to propagate the state, incorporate the measurement updates, and react to control inputs were trained. In addition, the covariances were designed as constants during the entire estimation. This approach can estimate the state better than a distinct prediction and update model, especially when large-scale training data are insufficient, as demonstrated in the experiment section in the present study.

The dynamic regression of KF parameters was put forward by [7] who adopted support vector regression (SVR) to estimate a linear state transition function, jointly with which the predicted NCM was estimated. The SVR-based system can deal with time-variant systems and outperforms manually tuned KF models in object tracking. As opposed to the model adopted here, the measurement noise covariances (MNCs) are kept constant and the transition function is modeled as a matrix multiplication. In this case, it can only estimate the linear motion models, while the model employed in the present study can estimate the nonlinear transition functions based on all previous state observations.

Reference [25] focused on integrating a one-shot estimation as a measurement into a KF model, which required a prediction of the MNC. They demonstrated that the integrated model exhibited superior performance by comparing it with two other models. In contrast, the model designed in the present work undertakes the measurement updates as a black-box system and automatically estimates the MNC, so that they can be combined with the current one-shot estimators.

Previous work has extensively investigated the temporal regularization for bit-pose estimation, and priority attention has been given to works that focus on implicit regularization schemes and that explicitly use a learning-based KF structure to infer temporal coherence. In contrast to other models, the proposed model introduces the LSTM-KF, which mitigates the modeler-induced influence on specifying motion and noise models a priori, while allowing rich models to learn from data that are extremely difficult to write down explicitly. An extensive series of experiments reveals that the LSTM-KF outperforms both the stand-alone KF and LSTM in terms of temporal regularization.

## 3. Materials and Methods

### 3.1. Overall Technical Architecture

The videos in the research area were acquired using Prometheus 230 intelligent UAVs (Chengdu Bobei Technology Co., Ltd., Chengdu, China), as shown in Figure 1. An Intel RealSense D435i stereo camera was selected for the system to acquire sensing and depth data because its features include light weight, wide field of view (FoV), high depth accuracy, and good stability. Furthermore, a powerful graphics processing unit (GPU) was employed for the embedded systems, and the NVIDIA Jetson AGX Xavier embedded platform was selected to process the deep-learning-based algorithms. A flight controller (Pixhawk 4 (PX4)) was deployed and communicated with the MAVROS package, which was connected to the planner node.

As illustrated in Figure 2, the designed system consists of (1) a perception module, (2) an object-tracking algorithm, (3) a UAV maneuver, and (4) a ground station visualization module. In brief, the UAV system perceives the red-green-blue (RGB) images and the depth data first, and the drone recognizes the Procapra przewalskii with YOLOX (a deep-learning-based detector). Next, the 2D bounding boxes are fused with the depth measurement to estimate the 3D pose of the Procapra przewalskii. Finally, the improved LSTM-KF model proposed here is integrated to assist in predicting the motion of the Procapra przewalskii. During this, the visualization user interface is included.

### 3.2. Dataset Establishment and Training

Precepting an object in a 3D world is essential for detecting and tracking an object. A deep-learning-based detector is employed here to generate related 2D information and perform 3D stereo reconstruction. This is very challenging because the object may move fast (e.g., running), the training data are low, the detection accuracy is not high, and the position of the object is changing continuously. As mentioned above, the YOLOX algorithm is selected as the baseline model to more accurately detect and track the Procapra przewalskii. Its structural framework is illustrated in Figure 3.

By referring to YOLOV3 and Darknet53, the YOLOX model adopts the structural architecture and spatial pyramid poling (SPP) layer of the latter. In addition, the model is equipped with several high-performance detectors.

In August 2022, we went to the research area (Qinghai Lake) (Figure 4a) to acquire the UAV data and verify the actual flight. There were 40 flights in five days, and each flight lasted about half an hour. The average flight height was about 100 m, and the aerial photography coverage area reached 9744 km^2^. Figure 4b shows the flight landing sites. With the captured videos, an object-tracking database was established to identify the moving Procapra przewalskii, match them in different frames, and track their motions.

A total of 6 video sequence databases, which were composed of 3 training databases and 3 test databases, are marked. The data were divided into a training set, a test set, and a verification set at a ratio of 3:2:1 (The Appendix A is available at link https://pan.baidu.com/s/1vEYdFFTKUE9Z9cC67lCH_Q?pwd=56vx). There were three major motions for Procapra przewalskii, which are standing, walking, and running, as displayed in Figure 5a–c (male), Figure 5d–f (female) and Figure 5g–i (young). The database was trained based on the YOLOX model by adjusting the weight ratio, confidence threshold, intersection over union (IoU) threshold of nms, and activation function. In this way, a stable and accurate model was obtained.

### 3.3. 3D Pose Estimation

Herein, the predicted bounding box was saved as *S_ROI_*, and the 3D pose of the object was recovered and dynamically tracked by the object coordinates on the 2D frame based on the depth information obtained from the stereo camera. In addition, an interior rectangle Si was firstly generated by contracting *S_ROI_* with a scaling factor θ, as computed in Equations (1) and (2). In the equations below, *S_i_* is the predicted bounding box; θ refers to the scaling factor to adjust the size of the bounding box; $c_{x}$ and $c_{y}$ are the coordinates of the bounding box; and *w* and *h* represent the width and height of the bounding box, respectively.
(1)$$S_{ROI}=[c_{x}\;c_{y}\;w\;h]$$
(2)$$S_{i}=[c_{x}\;c_{y}\;\theta w\;\theta h]$$

*S_i_*, as displayed in Figure 6b, serves as the *ROI* to obtain the depth information. The unfilled pixels are filtered out from the depth image captured by the stereo camera, and the remaining depth data in *S_i_* are averaged as *S*, which is assumed as the distance between the observer and the target object.

Then, with the boundary box coordination, coordination transformation was performed to obtain the relative attitude of the camera and the global attitude in the world frame. Frame transformation was carried out according to Equations (3) and (4) below:(3)$$S{\left[uv\right]}^{T}=K\cdot \left[\begin{array}{c} X_{i}^{C}\\ 1 \end{array}\right]$$
(4)$$\left[\begin{array}{c} X_{i}^{W}\\ 1 \end{array}\right]=T_{B}^{W}T_{C}^{B}\left[\begin{array}{c} X_{i}^{C}\\ 1 \end{array}\right],T_{C}^{W}T_{C}^{B}\in SO(3)$$

In the above equations, *u* and *v* are the pixel coordinates of $S_{t}$; *K* is the inherent matrix of the local camera and $X_{i}^{C}$ is the object pose vector in camera frame; and $X_{i}^{W}$ refers to the object pose vector in the world frame. Specifically, the transformation matrix can be calculated using Equations (5) and (6).
(5)$$T_{C}^{B}=\left[\begin{array}{cccc} 0 & 0 & 1 & 0\\ -1 & 0 & 0 & 0\\ 0 & -1 & 0 & 0\\ 0 & 0 & 0 & 1 \end{array}\right]$$
(6)$$T_{B}^{W}=\left[\begin{array}{cccc} r_{11} & r_{12} & r_{13} & o_{x}\\ r_{12} & r_{22} & r_{23} & o_{y}\\ r_{31} & r_{32} & r_{33} & o_{z}\\ 0 & 0 & 0 & 1 \end{array}\right]$$
where $r_{i\;j}$ is an element in the observer pose rotation matrix; $o_{x}$, $o_{y}$, and $o_{z}$ denote the position of the observer (UAV) relative to the world frame; and $T_{C}^{B}$ and $T_{B}^{W}$ are the pose transformation matrices. Rotation of the coordinate system is usually represented by a rotation matrix or a quaternion representation.

### 3.4. Tracking Based on the Improved LSTM-KF Model

In this study, the YOLOX algorithm is employed because it can balance speed and accuracy. Dynamic states of the target Procapra przewalskii and the quadrotor reduce the robustness of the pose estimation based on the descriptions in Section 3.3. The target Procapra przewalskii could not always be captured in the FoV during a surveillance as false positive or negative results may be found. In addition, partial or full occlusion might occur but not often. To address the above issues, the KF model is utilized to enhance tracking, but it requires the specification of a motion model and a measurement model in advance, which increases the burden on the modeler.

#### 3.4.1. Model Structure and Prediction Steps

As introduced above, an improved LSTM-KF model is proposed in the current study, which is a time regularization model for attitude estimators. Its main idea is to use the KFs without specifying a linear conversion function or fixed process and measuring the covariance matrixes *Q* and *R*.

The network of the standard LSTM (Figure 7) exhibits memory units, forgetting gates (*f_t_*), input gates (*i_t_*), and output gates (*O_t_*). Some information of cell state *C_t−_*_1_ is retained in the current cell state *C_t_*, and the amount of retained information is determined by *f_t_*, as given in Equation (7). Meanwhile, *i_t_* and *O_t_* can be calculated with Equations (8) and (9), respectively.
(7)$$f_{t}=\sigma (W_{f}\cdot [h_{t-1},x_{t}]+b_{f})$$
(8)$$\begin{array}{l} i_{t}=\sigma (W_{i}\cdot [h_{t-1},x_{t}]+b_{i})\\ C_{t}=\tanh(W_{c}\cdot [h_{t-1},x_{t}]+b_{c}) \end{array}$$
(9)$$\begin{array}{l} O_{t}=\sigma (W_{o}\cdot [h_{t-1},x_{t-1}]+b_{o})\\ h_{t}=O_{t}\cdot \tanh(C_{t}) \end{array}$$

Here, *O_t_* on the standard LSTM network is modified as follows:(10)$$Y_{t}=V_{t}C_{t}+\sum_{n=0}^{t-1}W_{nt}X_{n}$$

In Equation (10) above, {*X_0_*, *X_1_*, …, *X_t−1_*} and {*Y_t_*, *Y_t+1_*, …, *Y_t+n_*} are the input and output of the LSTM network, respectively; {*W_0(t)_*, *W_t_*, …, *W_(t−1)t_*} and {*W_0(t+1)_*, *W_t(t+1)_*, … *W_(t−1)(t+1)_*} represent the direct weights of the input and output, respectively; *C* refers to the current state of the LSTM network; and *V* is the coefficient.

KF is an optimal state estimator under the assumptions of linear and Gaussian noise. Specifically, if the state and the measurement state are expressed as $y_{t}$ and $z_{t}$, respectively, the hypothetical model here can be expressed in Equations (11) and (12).
(11)$$y_{t}=Ay_{t-1}+w,w\to N(0,R)$$
(12)$$Z_{t}=Hy_{t}+v,v\to N(0,R)$$

Because the incoming measurements are noisy estimates of potential states and *H = I* in Equation (11), Equations (11) and (12) can be modified to Equations (13) and (14), respectively, which are the basic models of LSTM-KF. In the equations below, *Z_t_* denotes the model measurement state; $W_{t}$ is the weight at moment *t*; $Q_{t}$ and $R_{t}$ are covariance matrices; and *f* is a nonlinear transfer function.
(13)$$y_{t}=f(y_{t-1})+w_{t},w_{t}\to N(0,Q_{t})$$
(14)$$Z_{t}=y_{t}+v_{t},v\to N(0,R_{t})$$

The prediction step can be defined by Equations (15) and (16).
(15)$${\hat{y}}_{t}^{\prime }=f({\hat{y}}_{t-1})$$
(16)$${\hat{P}}^{\prime }=F{\hat{P}}_{t-1}F^{T}+{\hat{Q}}_{t}$$
where *f* is modeled by an LSTM module, *F* is the Jacobian matrix of *f* relative to ${\hat{y}}_{t-1}$, and ${\hat{Q}}_{t}$ is the output of the second LSTM model. Thus, the updating steps are specified in Equations (17)–(19).
(17)$$K_{t}={\hat{P}}_{t}{}^{\prime }{\left({\hat{P}}_{t}+{\hat{R}}_{t}\right)}^{-1}$$
(18)$${\hat{y}}_{t}={\hat{y}}_{t}^{\prime }+K_{t}({\hat{z}}_{t}-{\hat{y}}_{t}^{\prime })$$
(19)$${\hat{P}}_{t}=(I-K_{t}){\hat{P}}_{t}{}^{,}$$
where ${\hat{R}}_{t}$ is the output of the third LSTM module and ${\hat{z}}_{t}$ refers to the observed measurement at time t. Next, these LSTM modules are described in detail.

#### 3.4.2. Architecture and Loss Function

In this paper, LSTM_f_, LSTM_Q_, and LSTM_R_ are selected to represent the three LSTM modules of *f*, ${\hat{Q}}_{t}$, and ${\hat{R}}_{t}$, respectively. LSTM_f_ is composed of three stacked layers (1024 hidden cells in each) and three fully connected (FC) layers (with 1024, 1024, and 48 hidden cells). Similarly, standard LSTM is built as LSTM_f_, but $O_{t}$ of the LSTM_f_ is modified to connect to *i_t_*. In addition, ReLU nonlinearity is introduced to all FC layer activations except the last, and each LSTM layer is followed by a lost layer with a retention probability of 0.7. LSTM_Q_ and LSTM_R_ follow with 256 hidden unit monolayer frameworks and 48 hidden units. Meanwhile, $O_{t}$ connects with *i_t_* to prevent the video time sequence of the agency and the Procapra przewalskii training to avoid overfitting. Figure 8 shows each module of the LSTM-KF model and Figure 9 displays an overview of the system.

At each *t*, taking ${\hat{y}}_{t-1}$ as an input, LSTM_f_ generates the intermediate state ${\hat{y}}_{t}^{\prime }$ without depending on the current measurement; LSTM_Q_ takes ${\hat{y}}_{t}^{\prime }$ and ${\hat{Q}}_{t}$ as the input and output, respectively, and estimates the process covariance; and taking *Z_t_* and ${\hat{R}}_{t}$ as an input and output, respectively, LSTM_R_ only estimates the measured covariance. Finally, ${\hat{y}}_{t}^{\prime }$ and *Z*_t_, along with the covariance estimates made here, are fed into a standard KF (Equations (16)–(19)), ultimately yielding a new prediction ${\hat{y}}_{t}$. Moreover, *Q* and *R* are restricted to diagonal and positive definite by indexing the output of the LSTM_Q_ and LSTM_R_ modules in this study.

Preliminarily, the standard Euclidean loss summation is applied throughout the entire process, but the LSTM_f_ module fails to learn reasonable mapping. Therefore, the loss function is introduced with a term to enhance the gradient flow to the LSTM_f_ module in the current study. Equation (20) expresses the specific loss function.
(20)$$L(\theta )=\frac{1}{T}\sum_{t=1}^{T}\left||y_{t}-{\hat{y}}_{t}(\theta \right)|{|}^{2}+\lambda ||y_{t}-{\hat{y}}_{t}^{\prime }(\theta )|{|}^{2}$$

#### 3.4.3. Optimization of Parameters

All parameters *θ* in the loss function are optimized to minimize the loss given by Equation (20) regarding all free parameters in the model applied in this work, which is the connection of the ownership weight matrix and bias from all three LSTM modules, which are a combination of the LSTM layer and the linear layer (Figure 8).

The LSTM-KF model can achieve end-to-end training, and the gradient can be obtained through the time backpropagation algorithm [26]. All the computations and states in the model are presented by a single data flow graph, so that communication between the sub-computations can be displayed, which is conductive to the parallel execution of independent computations to obtain the gradient as soon as possible.

In addition, the Adam-based optimizer [27] is introduced for training iteration to update the gradient, ensuring high stability and high predictability of the model. Meanwhile, the update rules are extended based on the *L^2^* norm to those based on the *L^p^* norm. A large *p* results in numerical instability of these variants. However, in a special case of *p → ∞*, the algorithm is simple but stable, which can be calculated with Equation (21).
(21)$$v_{t}=\beta_{2}^{p}v_{t-1}+(1-\beta_{2}^{p})|g_{t}{|}^{p}$$

With the *L^p^* norm, the step size at time *t* is supposed to be inversely proportional to *v_t_^1/p^*, and then Equation (21) can be modified to Equation (22).
(22)$$v_{t}=(1-\beta_{2}^{p})\sum_{i=1}^{t}\beta_{2}^{p(t-i)}\cdot |g_{i}{|}^{p}$$

It should be noted that the attenuation term is equivalently parameterized here to $\beta_{2}^{p}$ instead of *β^2^*. If *p → ∞* and *v_t_ = lim_p_
_→ ∞_(v_t_)^1/p^* are defined, Equations (23)–(26) can be obtained. This corresponds to a very simple recursive equation (Equation (27)).
(23)$$v_{t}=\lim_{p\to \infty }{\left(v_{t}\right)}^{1/p}=\lim_{p\to \infty }{\left((1-\beta_{2}^{p})\sum_{i=1}^{t}\beta_{2}^{p(t-i)}\cdot |g_{i}{|}^{p}\right)}^{1/p}$$
(24)$$=\lim_{p\to \infty }{\left(1-\beta_{2}^{p}\right)}^{1/p}{\left(\sum_{i=1}^{t}\beta_{2}^{p(t-i)}\cdot |g_{i}{|}^{p}\right)}^{1/p}$$
(25)$$=\lim_{p\to \infty }{\left(\sum_{i=1}^{t}\left(\beta_{2}^{\left(t-i\right)}\cdot |g_{i}{|}^{p}\right)\right)}^{1/p}$$
(26)$$=\max(\beta_{2}^{t-1}|g_{1}|,\beta_{2}^{t-2}|g_{2}|,\ldots ,\beta_{2}|g_{t-1}|,|g_{t}|)$$
(27)$$v_{t}=\max(\beta_{2}\cdot v_{t-1},|g_{t}|)$$

The initial value is *v_0_ = 0*. Note that conveniently, the initialization bias does not necessarily need to be corrected. The improved Adam-based optimizer is simpler than the original and is easier for gradient updating.

## 4. Result

The performance of the trained model was assessed on a Jetson AGX Xavier onboard computer, where the coupled detection head of the original YOLO model was replaced with the decoupled head, which greatly accelerated training convergence. As mentioned, the robustness of the proposed model was observed on a streaming video with various techniques for quantitative analysis Lastly, several intensive flight tests were performed on a self-assembled quadrotor platform to evaluate the overall performance.

### 4.1. Detection Effect of the YOLOX Model

The YOLOX model outputs a predictive bounding box that classifies detected objects and marks their locations, which plays a key role in subsequent pose estimation using the UAV. The model training lasted for 1000 iterations, during which time the loss did not degrade. Because the accuracy of the model is affected by different neural network resolutions, the YOLOX model was trained with different resolutions (i.e., 416 × 416, 512 × 512, 640 × 640) to evaluate the best performance. The four input resolutions are compared in Table 1.

Furthermore, the surveillance performed by UAVs was realized based on real-time perception solutions, which focus more on object detection and tracking. In this case, the detection speed and accuracy had to be balanced to ensure consistent detection and tracking, in which delay can be neglected and accuracy is high enough. Thus, the YOLOX and YOYLOv4 models of the same network resolution were compared to examine their accuracy and speeds.

After training, the model performance in detecting target Procapra przewalskii on real-time videos captured was evaluated on an Intel RealSense D435i stereo camera. The trained model was proven to be robust under various environments and exhibits low false positives and negatives. Procapra przewalskii tracking was then successively assessed after assuring the validity of the model.

### 4.2. Tracking Performance on Target

In this study, the LSTM-KF model was employed to track and identify the behaviors and gestures of the video sequence dataset of Procapra przewalskii. Six object-tracking sequences were comprehensively generated from the Procapra przewalskii dataset, and the 6-DOF ground realistic attitude was available. The LSTM-KF model was trained at $2\times 10^{-5}$, decaying by 0.95 from the second period. Before training, gradients of 100 time steps were propagated using a truncated backpropagation time.

However, for a single-layer LSTM with 16 hidden units, batch size is set to 2 and the learning rate is designed as $5\times 10^{-4}$. After the model is trained for 120 periods, the gradient is propagated again for 10-time steps in the same way. It is the same case for the standard LSTM method evaluated in this work.

The tracking algorithm can be evaluated by employing the successive frames of depth frame sequence tracking through the 3D CAD model of 3D pose. Therefore, all the task methods are compared here to obtain a target-tracking method which is superior to the existing methods. Table 2 displays the results of tracking recognition under the scenario.

### 4.3. Verification of Field Tracking Flight

To integrate a perception to reaction and evaluate the surveillance system, four sites (the red points in Figure 4b) were selected to verify the tracking effect of the UAV on Procapra przewalskii. The parameter settings for the flight test are shown in Table 3.

Due to the complexity of the algorithm and the uncertainty of the research site, simulation verification and parameter adjustment of the algorithm are necessary before actual flight. Prometheus is an open-source autonomous drone software platform with seamless switching from simulation to real operation (https://wiki.amovlab.com/public/prometheus-wiki/Prometheus-%E8%87%AA%E4%B8%BB%E6%97%A0%E4%BA%BA%E6%9C%BA%E5%BC%80%E6%BA%90%E9%A1%B9%E7%9B%AE/Prometheus-%E8%87%AA%E4%B8%BB%E6%97%A0%E4%BA%BA%E6%9C%BA%E5%BC%80%E6%BA%90%E9%A1%B9%E7%9B%AE.html, accessed on 12 August 2021), emergency safety protection mechanisms, interactive ground stations, unified interfaces, and code specifications. Figure 10 shows the validation scenario of the Prometheus simulation system, which simulated the LSTM-KF model proposed in this paper to track Procapra przewalskii. The field verification results are displayed in Figure 11.

Additionally, the estimated dynamic position is compared with the ground truth of the tracked Procapra przewalskii. Figure 12 and Figure 13 demonstrate that the system can basically track the poses of Procapra przewalskii in 3D space. Regardless of jittering and occasional drift, the Procapra przewalskii can be relocated accurately after several frames. In addition, the figure shows that error is basically within 0.5 m in all axes of the world frame.

In addition, the root-mean-square error (*RMSE*) and mean absolute error (*MAE*) (defined in Equations (28) and (29), respectively) were calculated, as presented in Table 4. Here, *RMSE* indicates the degree of prediction error generated by the model, and a large error results in heavier weights. *MAE* reflects the error between the predicted and actual values, and a smaller *MAE* corresponds to a better model performance. $y_{i}$ and ${\hat{y}}_{i}$ in Equations (28) and (29) are the true value and the predicted value, respectively.
(28)$$RMSE=\sqrt{\frac{1}{n}\sum_{i=1}^{n}{\left(y_{i}-\bar{y_{i}}\right)}^{2}}$$
(29)$$MAE=\frac{1}{n}\sum_{i=1}^{n}\left|y_{i}-\bar{y_{i}}\right|$$

In addition, during monitoring, the distance between the UAV and the target Procapra przewalskii constantly changed, so it is necessary to further analyze the accuracy at different distances. As shown in Table 5, the performance of the proposed method remained basically stable regardless of the distance between the UAV and the target Procapra przewalskii.

## 5. Discussion

It should be highlighted that if the input resolution is too large, the best possible mAP increases, but the training and detection speed are indeed negatively affected. Therefore, the higher input resolution is not trained in this work, because the currently obtained speed and accuracy at 640 × 640 are acceptable, and the mAP and the union threshold intersection are 88.67% and 0.50 (AP50), respectively. Unfortunately, the fps is not as fast as the resolution of 512 × 512. Meanwhile, it was found that the mAP of the YOLOX model was lower in contrast to the YOLOv4 model, but the fps was higher. In consideration of the greater significance of fps in real-time predictions, the YOLOX model with a higher fps was selected to balance the accuracy and speed.

Table 2 clearly displays that the motion models that do not investigate the training data, i.e., Kalman Vel Kalman Acc and EMA, are not meaningfully improved for translational estimation and rotation. However, the improved LSTM-KF model proposed in this paper performs better in predicting the target position (0.82 mm) with a mean error of 61.26%, which is higher than the original estimate and better than the results in [28], using the KF algorithm alone for target tracking, and exhibits a lower average error. In addition, the LSTM-KF model greatly improves the original measurements with all actions outperforming the standard LSTM by an average of 14% compared to the state-of-the-art method. In contrast, the standard LSTM method estimates the position and rotation with such a large error that they fail to meet the requirements.

As shown in Table 3 and Table 4, the 3D object pose-estimation systems focused on by other scholars [29,30] highlight the objects in static states, while the model applied in this paper exhibited higher errors in estimating the dynamic position of Przewalski’s Tibetan antelope in real time. However, it possesses better robustness and is also more accurate than the model in [28] that uses the KF algorithm alone for 3D pose estimation of the target. In addition, it mitigates the influence of the modeler on the a priori specified motion and noise models. The model for 3D pose estimation of dynamic targets using an improved spatio-temporal context algorithm in comparison to that in [31] exhibits higher accuracy, a fast network convergence, and fewer impacts from measurement noise. Overall, the proposed LSTM-KF possesses a better performance in pose estimation than the sole use of the KF algorithm and improved spatio-temporal context algorithm. If the target animal, Przewalski’s Tibetan antelope, moves suddenly, redundant overshot periods follow, slightly affecting the overall performance of the model. Thus, it further proves that the proposed model can be better applied to autonomous UAV monitoring systems in a real-time and manipulable manner.

## 6. Conclusions

In this paper, a deep-learning-based model was employed to build an autonomous UAV tracking system to help monitor Procapra przewalskii. The LSTM-KF model is proposed and applied to track the target, and the YOLOX model is employed to identify the target. Meanwhile, they were combined to estimate the pose of the protozoa in 3D images, thus improving the performance of object tracking. In addition, the three different standard LSTM networks modeled in the LSTM-KF model are optimized by modifying and connecting the computation of *Q_t_*. During the training iterations of the LSTM-KF model, Adam as an optimizer is improved by extending the *L^2^* criterion-based update rule to an *L^p^* criterion-based rule. The results show that the improved LSTM-KF model can achieve the best result (0.82 mm) in predicting the target position with an average error of 61.26%, which is higher than the original estimate, significantly improving the accuracy of the measurement results. In addition, the YOLOX model exhibits an mAP of 93.25% and an FPS of 13.56 on images with 640 × 640 resolution, which are higher than those of the YOLOv4 model. Overall, the proposed improved LSTM-KF model is robust, valid, and reliable for animal tracking, recognition, and pose estimation.

However, this paper is subject to several shortcomings for Procapra przewalskii tracking. For example, when UAVs are applied to track dense herds of Procapra przewalskii, accuracy is decreased if Procapra przewalskii individuals occlude each other. Based on this, we are trying to solve this problem in future research using algorithms based on depth sorting. In addition, it is believed that a visual servo controller can be designed to control the UAV to explore the environment or avoid obstacles using only one camera, ensuring that the tracked object is always in the field of view of the camera.

## Figures and Tables

**Figure 1 sensors-23-03948-f001:**
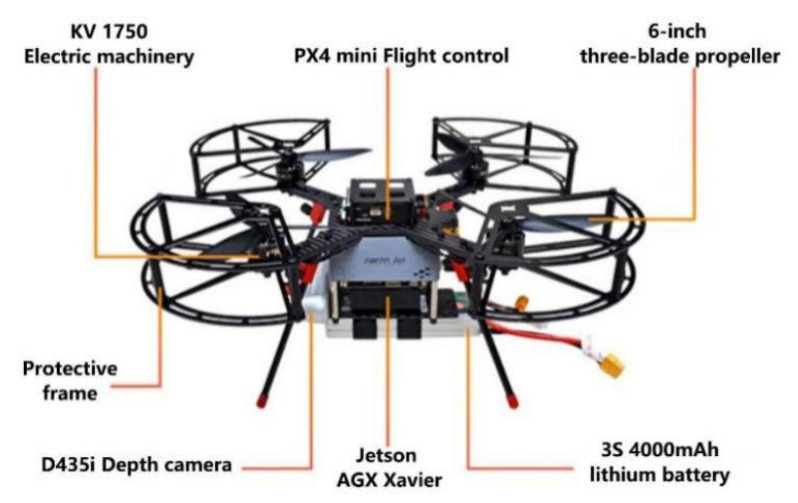
Prototype of the proposed autonomous object-tracking system.

**Figure 2 sensors-23-03948-f002:**
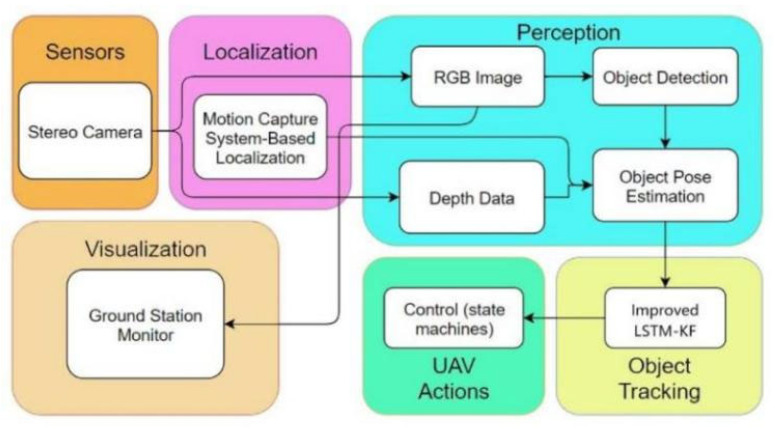
Software architecture of the system.

**Figure 3 sensors-23-03948-f003:**
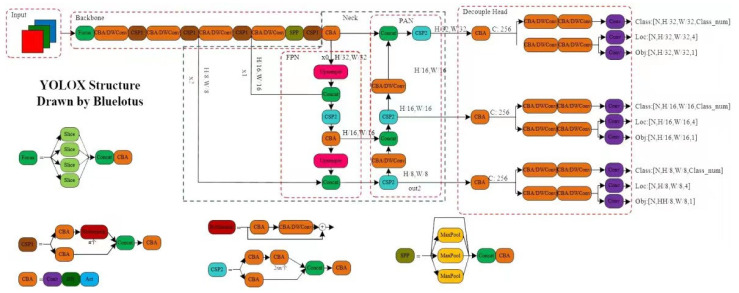
Structural framework of the YOLOX algorithm.

**Figure 4 sensors-23-03948-f004:**
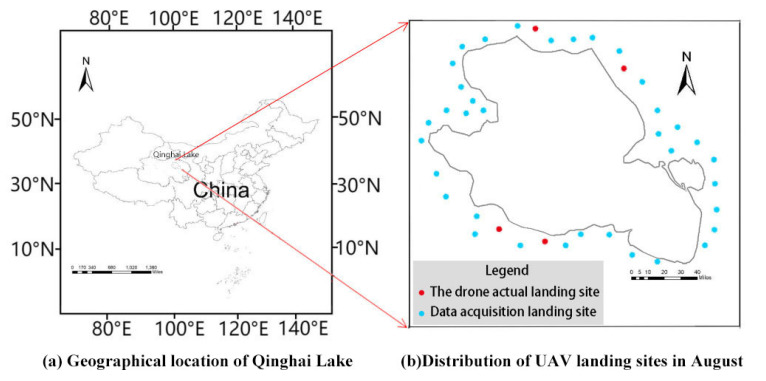
Research area: (**a**) geographical location of Qinghai Lake; (**b**) distribution of UAV landing sites in August.

**Figure 5 sensors-23-03948-f005:**
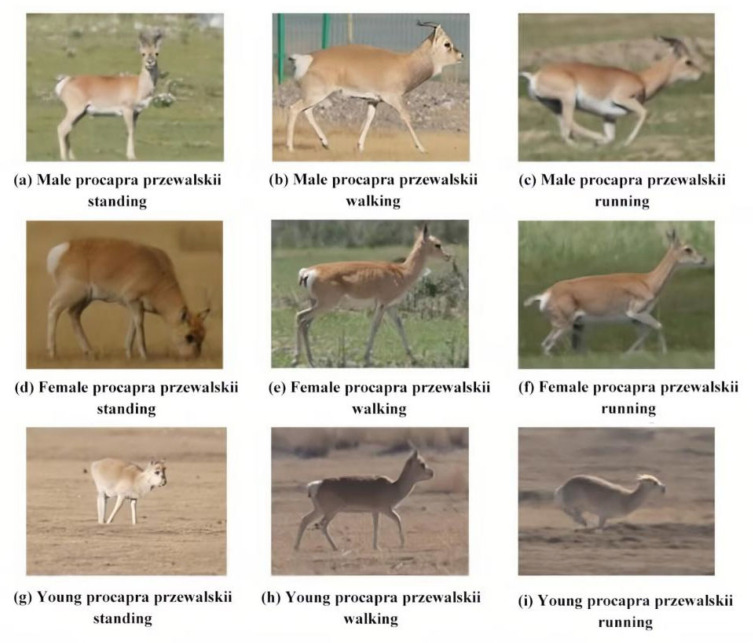
Different motions of male, female, and young Procapra przewalskii.

**Figure 6 sensors-23-03948-f006:**
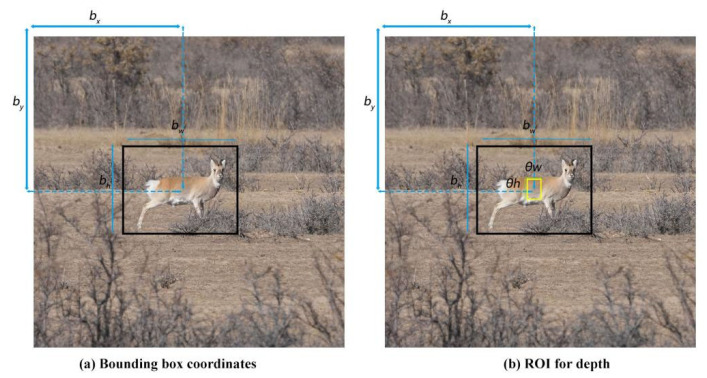
(**a**) Bounding box coordinates; (**b**) *ROI* for depth.

**Figure 7 sensors-23-03948-f007:**
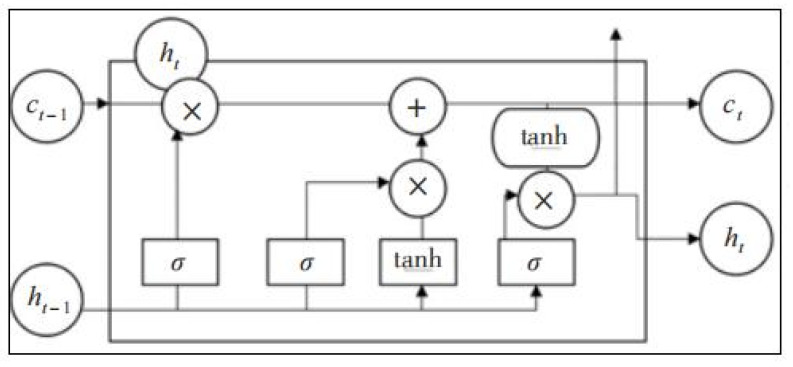
Network structure of the standard LSTM.

**Figure 8 sensors-23-03948-f008:**
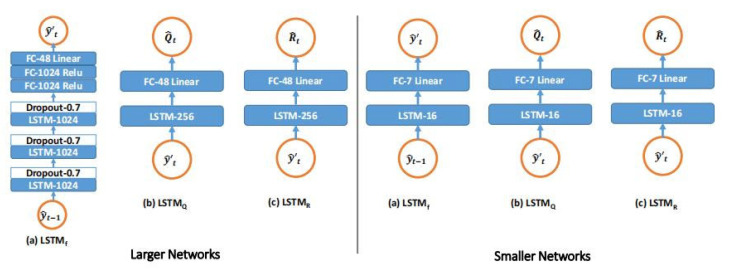
Structure of the LSTM-KF model.

**Figure 9 sensors-23-03948-f009:**
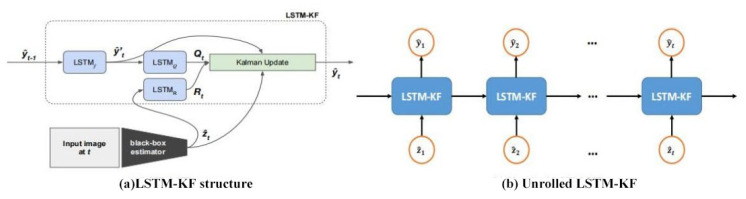
Overview of the LSTM-KF model: (**a**) a high-level depiction of the architecture which uses three LSTM modules to predict the internals of the KF; (**b**) LSTM-KF unrolled over time, which can be trained end to end with backpropagation.

**Figure 10 sensors-23-03948-f010:**
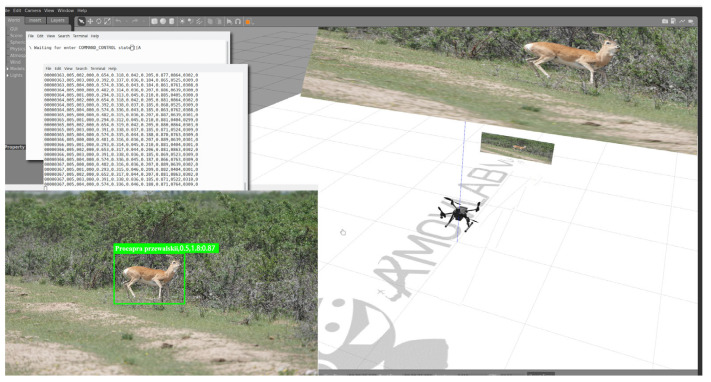
Verification scenario for the tracking algorithm of Procapra przewalskii based on the Prometheus simulation system.

**Figure 11 sensors-23-03948-f011:**
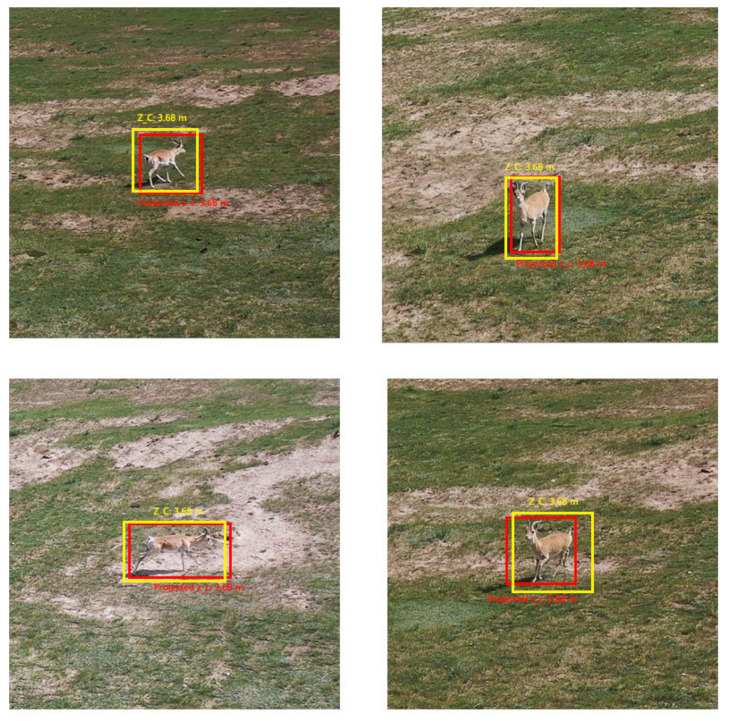
First-person views of the UAV system during the Procapra przewalskii track (the yellow and the red bounding boxes are the detected and predicted states of the object, respectively).

**Figure 12 sensors-23-03948-f012:**
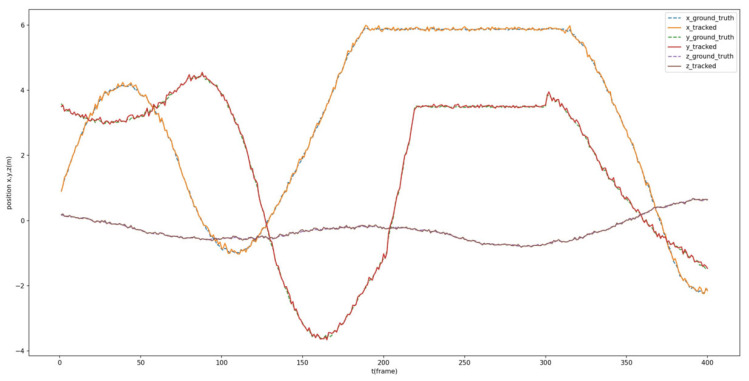
Comparison between the estimated position and ground truth of Procapra przewalskii.

**Figure 13 sensors-23-03948-f013:**
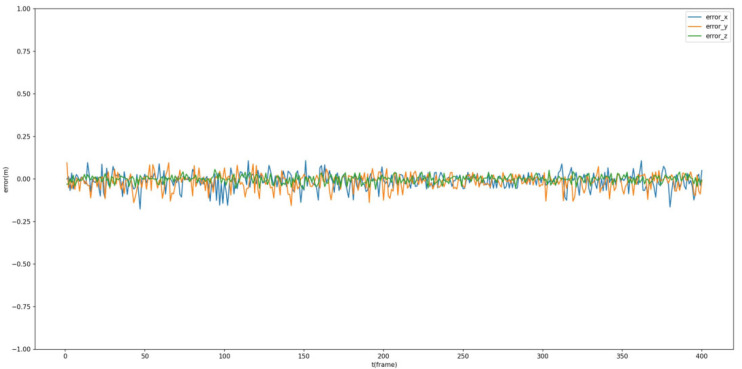
Error throughout the mission time.

**Table 1 sensors-23-03948-t001:** Performances of YOLOv4-Tiny and YOLOX with respect to different resolutions.

Method	Backbone	Size	mAP@0.5 (AP50)	FPS
YOLOv4	CSPDarknet-53	416 × 416 512 × 512 640 × 640	83.19% 85.87% 88.67%	6.12 6.24 6.02
YOLOX	Darknet-53	640 × 640	93.25%	13.56

**Table 2 sensors-23-03948-t002:** Effects of temporal regularization on object-tracking estimations of Procapra przewalskii. The errors in translation are denoted as [mm].

	Male Trans	Female Trans	Young Trans	Mean Trans
D.J. Tan et al.	1.58	2.55	3.85	2.66
+Kalm an Vel.al	1.49	2.42	3.34	2.42
+Kalm an Acc.	1.49	2.41	3.32	2.41
+EMA	1.59	2.56	3.87	2.67
+Std. LSTM	40.98	45.98	50.15	45.7
LSTM-KF (ous)	0.58	0.67	1.22	0.82

**Table 3 sensors-23-03948-t003:** Defined parameters for flight test.

Parameters	Value
*θ*	12 deg
V*_θmax_*	45 deg/s
V*_Zmax_*	2 m/s
V*_Xmax_*	2 m/s
R*_safe_*	5 m
R*_sur_*	30 m

**Table 4 sensors-23-03948-t004:** Calculated *RSME* and *MAE* for position estimation of the dynamic Procapra przewalskii.

Error Evaluation	X(m)	Y(m)	Z(m)
RMSE MSE	0.0477 0.0022	0.0478 0.0023	0.0221 0.0005

**Table 5 sensors-23-03948-t005:** Calculated *RSME* and *MAE* of position estimation of dynamic Procapra przewalskii with different object distances.

Object Distance	10–30			30–50		
Error Evaluation	X(m)	Y(m)	Z(m)	X(m)	Y(m)	Z(m)
RMSEMSE	0.0477 0.0022	0.0478 0.0023	0.0221 0.0005	0.0773 0.0059	0.0805 0.0648	0.0656 0.0431

## Data Availability

The supplemental data set for this study is available in Appendix A.

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
