# Peer review of "Procapra Przewalskii Tracking Autonomous Unmanned Aerial Vehicle Based on Improved Long and Short-Term Memory Kalman Filters"

_sensors, 2023, doi:10.3390/s23083948_

Round 1

Reviewer 1 Report

This manuscript presents an autonomous unmanned aerial vehicle tracking system based on an improved long and short-term memory Kalman filter model. On the whole, the research content of this manuscript is relatively meaningful, but there are also some problems that need to be modified.

1. The content of this manuscript lacks some practical content, such as experimental environments, simulation validation, etc. The author mainly uses simple language to describe and provide tables, lacking practical evidence, which can easily lead to readers' doubts about the understanding of the content of this manuscript.

2. In the last paragraph of Section 1, it is necessary to clearly give the section arrangement of this manuscript to facilitate the readers to understand this manuscript.

3. Please further refine the language of the manuscript, some of which are not well explained. For example, “the Equations (11) and (12) can be revised into Equations (13) and (14), ...” should be revised to “Equations (11) and (12) can be revised into Equations (13) and (14), ...”

4. Generally, the definite article is not given in the graphic of manuscript. For example, “Figure 11. The ground truth and estimated position of the tracked procapra przewalskii for comparison”. “the” should be deleted.

5. The variable should be italicized, and this issue is evident in this manuscript.

6. Some of the figures are not clear, making it difficult to see the text and information in the figures clearly.

7. Many variables in equations are not explained in detail.

8. The authors of this manuscript were careless and made many low-level errors. It is recommended to carefully check the full text. For example

The detailed results are displayed in Fig. 10.

Its structural framework is illustrated in Fig. 3.

Please confirm whether the Dardknet-53 in Table 1 are correct.

9. In nature, herbivores generally engage in social activities, but this manuscript only uses a single animal to verify the proposed method, which is relatively simple. It is recommended to add identification verification for group activities of multiple animals to verify the broad adaptability and robustness of the methods proposed in this manuscript.

10. In Figure 11, a lot of information is displayed overlapping and confusing to the reader. It is recommended that key information can be partially enlarged for display.

11. In Figure 12, the curve fluctuates greatly, and the reason for this phenomenon is not given in this manuscript. Since the curve fluctuates greatly, how to verify the method proposed in this manuscript. At the same time, it is recommended to use multilinear and multi-color representations of the curves to improve the reader's recognition. The key information can also be displayed in the form of partial enlargement.

12. The conclusion of this manuscript is too concise.

13. The font in the figure does not follow the format of Sensors.

Author Response

We have greatly improved the language of the paper. Please refer to the attachment for other questions.

Reviewer 2 Report

There are too many authors for the article from eigth different institutions. Maximun five people recommended.

The results seem adequate for a tracking algorithm that authors presents. It is recommended to aggregate more pictures of verification of field tracking flight.

Authors are recommended to improve the quality of images.

The link in appendix A did not work. Requires custom acces password. 

Authors are advised to be careful with this type of information that is included.

Author Response

We have answered the reviewer's questions in detail in the attachment.

Reviewer 3 Report

Authors contributions

The authors have proposed an autonomous unmanned aerial vehicle (UAV) tracking system based on an improved long and short-term memory Kalman filter model. The system can estimate the three-dimensional attitude and track the target object precisely without manual intervention. The results show that the improved LSTM-KF model could remarkably better the original measurement results.

 Have some reviewer notes:

Abstract. At the beginning, you have to define why tracking of the UAV is important. Next is to define the problem that you have to solve. Also, you have to show the results accuracy with values, not only by text description.

Introduction. The aim of this work is not clearly presented.

2.Related Works. At the end of this part you have to summarize what are the problems in the known literature and how you will solve them in your work.

Line 154. Prometheus 230 intelligent UAVs (Manufacturer, City, Country).

Line 159. NVIDIA Jetson AGX Xavier (Manufacturer, City, Country).

Line 160. Pixhawk 4 (PX4) (Manufacturer, City, Country).

Line 161. MAVROS package (Manufacturer, City, Country).

Equations (1) and (2). You have to describe the variables in these equations. Same for all equations that you present. More detailed description of used variables is needed.

Table 1. It is not clear what 100% precision means and what 83.19%, 85.87%... mean.

Equations (28) and (29). If it is possible, move these equations in Material and methods part.

Discussion part. You have to compare your results with minimum 3 other papers (more than 3 is appropriate).

Conclusion part. You have to present your results with accuracy values. How your work improves the known solutions in this study area? What are the limitations of your work? How your work can be continued? How your results can be implemented in the practice?

I have some suggestions:

Improve the presentation of your equations. Make more detailed description of study problem and your findings. Make more comparative analyses. These suggestions will improve your contribution.

Author Response

We have made the greatest improvement in the language of the paper and have answered the reviewers' questions one by one in the attachment.

Round 2

Reviewer 1 Report

According to the comments of the reviewer, the authors of manuscript have made detailed revisions to this manuscript, making the manuscript’s content more abundant and reasonable, and the language more fluent. The formula format meets the requirements of Sensors. Therefore, the reviewer believes that the manuscript can be acceptable.